# Interactive effects of developmental and adult nutrition on lifespan and fecundity in a genetically diverse *Drosophila* population

Andrew Jones, De'anne Donnell, Elizabeth G. King, Enoch Ng'oma *

Division of Biological Sciences, University of Missouri-Columbia, Missouri, United States of America

* ngomae@missouri.edu

## Abstract

Developmental conditions, including temperature, diet, and parasite exposure, can shape adult fitness phenotypes across species. While studies often examine the independent effects of early-life and adult conditions on life history traits, fewer have focused on their interactive effects, particularly in genetically diverse populations. Here, we investigate how larval and adult diet interact to influence lifespan and fecundity in a genetically diverse population of *Drosophila melanogaster*. We manipulated protein availability across larval and adult stages and found significant larval-adult diet interactions affecting both traits, though in different ways. Several key patterns emerged, including age- and sex-specific effects, survival differences in the post-median life phase, shifts in the timing of peak fecundity, and sustained egg production in older stages. Protein reduction increased male maximum lifespan and female lifetime fecundity, while lower adult protein intake delayed egg-laying by approximately two weeks, particularly in flies that also experienced low developmental protein. These findings underscore the importance of assessing life history traits dynamically over time, as interactions between developmental and adult environments may drive complex, non-additive effects that are not apparent in cross-sectional measurements. Considering both additive and interactive effects in diverse genetic backgrounds will be critical for understanding the evolutionary and ecological consequences of nutritional variation.

## Introduction

An organism's adult phenotype emerges from complex interactions between its ancestral, developmental, and current environmental conditions, including temperature [1–3], food availability [4–6], microbiotas [7,8], and parasite exposure [9]. Among these, nutritional conditions play a key role in shaping metabolic state dynamics that influence reproductive output and longevity across taxa, from yeast to primates [3,10–12]. Given that resource availability constrains energy allocation to competing

**Data availability statement:** All data, including lifespan records, egg counts, and original egg images were uploaded to and can be accessed from https://zenodo.org/records/10445786. Scripts and project notes to reproduce this analysis may be accessed at https://github.com/nochet/early_experience.

**Funding:** This work was supported by a NIH grant R01 GM117135 to EGK, and the University of Missouri startup fund to EN. There was no additional external funding received for this study.

**Competing interests:** The authors have declared that no competing interests exist.

fitness functions, an organism's reproductive schedule and survival are shaped by early-life and adult nutritional conditions. The extent to which developmental nutrition determines adult performance has been widely studied, with evidence suggesting that diet effects can persist and interact across life stages and influence survival and reproductive success [4,13,14]. For instance, poor maternal nutritional conditions can limit offspring development through inadequate and low-quality provisioning and care, with long-term consequences for offspring fitness [15].

The effects of early-life nutrition on adult fitness traits have been demonstrated in many taxa, including insects [3,9,16,17], birds [1,5,18,19], fish [20,21], and mammals [22]. Often, the effects of early life nutrition on adult reproduction have been studied in the context of capital breeders (those using stored resources gained early in life for reproduction) and income breeders (those using recently acquired resources in reproduction) [23]. While *Drosophila melanogaster* has typically been characterized as an income breeder because protein acquired at the larval stage is not used directly in eggs at the adult stage [24], diet manipulations during development have been shown to influence multiple fitness traits, with some traits (e.g., ovariole number, pupal mass, and wing length) responding more strongly to poor than rich diets [4]. Other traits, such as development time, responded to both extremes of nutritional variation, highlighting trait-specific sensitivity to environmental inputs [4,25]. Similar life stage-dependent responses have been observed in vertebrates that show indeterminate growth. For example, in the cichlid fish *Simochromis pleurospilus*, adult growth rate was dictated by current nutritional conditions, whereas reproductive traits were largely determined by juvenile diet, indicating potential trait-level decoupling of nutritional effects across life stages [20,21].

Early life conditions, including nutrient availability, can interact with individual genetic and/or epigenetic states and sex to shape adult fitness trajectories. For example, selection for adaptation to high- vs low-nutrient larval diet and early- vs late-life reproduction involved significant genetic interactions during experimental evolution in *D. melanogaster* [26]. Similar studies in *D. melanogaster* have shown that male genotype-by-diet interactions account for a significant proportion of the variation in male reproductive traits in isogenic lines [27]. Further, larval diet has been linked to decreased ribosome expression, altered transcription and translation negatively correlated with lifespan [28]. In the nematode *Caenorhabditis elegans*, transient developmental exposure to reactive oxygen species (ROS) was found to enhance stress resistance and prolong lifespan through interactions with ROS-sensitive epigenetic marks [29]. Furthermore, in mice, *in utero* nutritional deficits were shown to influence offspring growth by modifying epigenetic states at multicopy ribosomal DNA elements [30]. These studies, among others, highlight the role of early diet in shaping adaptive responses to nutritional variability.

Sexes can differ in their sensitivity to nutrition during development, leading to distinct physiological and fitness outcomes in adulthood. For example, a study in *Drosophila* found that mismatches between developmental and adult nutrition influenced female, but not male, reproductive success [31]. Survival in both sexes was primarily determined by adult diet, with longer lifespans on high-yeast adult food.

However, in females, developing on a high-yeast diet conferred additional benefits to lifespan and reproductive success, regardless of the adult diet consumed. In contrast, a high-yeast developmental diet was only beneficial for male lifespan when followed by low-yeast adult food. These patterns of sex-dependent dietary response are not unique to insects. In rodents, males show greater hyperphagia (overeating), whereas females are more inclined toward high-fat diets and are somewhat protected from rapid obesity and metabolic decline through higher energy expenditure [32]. Another study in mice suggested that female mice are relatively less sensitive to the metabolic improvements observed following dietary protein dilution compared to males, indicating sex-specific metabolic responses to diet composition [33]. In captive zebra finches, however, a recent study [18] could not find sex-specific environmental sensitivities, although female lifespan was shorter due to senescence, concluding that low food availability shortened lifespan only in individuals developed in harsh conditions. Together, these studies underscore the complexity of sex-specific nutritional effects, which can vary across species and environmental contexts emphasizing the need to consider both developmental and adult dietary influences on fitness outcomes.

Despite growing evidence that early-life nutrition shapes adult traits, relatively few studies have explored how inter-actions between developmental and adult diets affect fitness in genetically diverse populations. The genetic architecture underlying life history trade-offs may modulate responses to nutritional mismatches between life stages. In our prior work, dietary manipulation led to a 37.5% lifespan increase in a low-protein-treated *Drosophila* group relative to controls, highlighting the potential for diet-dependent longevity effects [34]. Building on this, we investigate how developmental and adult nutrition jointly influence lifespan and fecundity in outbred *Drosophila*, using a highly diverse genetic background derived from a multiparent recombinant inbred population [35–37]. By examining fitness outcomes across four dietary conditions varying in protein provision, we evaluate:

1. the combined effect of early-to-adult nutritional experience on lifespan and fecundity, and,

2. whether early-life nutrition exerts a stronger influence on adult fitness than later-life dietary conditions.

## Materials and methods

### Experimental population

We used an admixed *B* population of the *Drosophila* Synthetic Population Resource (DSPR) [35,36]. To generate the out-bred population, we inter-crossed 835 recombinant inbred lines (RILs) for five generations earlier [38]. From the outbred population, we maintained a single large cage (with >4,000 flies) on maintenance diet until we were ready to set up the present experiment in 2020.

### Study design

We used a 2 x 2 factorial design (Fig 1) in duplicate. Our treatments include two life cycle stages – larvae and adults; and two levels of environmental variation – low protein (L), and high protein (H) sources. These are designated as Larvae-High x Adult-High, Larvae-High x Adult-Low, Larvae-Low x Adult-High and Larvae-Low x Adult-Low, hereon, HH, HL, LH and LL. To obtain assayed individuals, eggs oviposited on media plates (i.e., polystyrene plate, 100 mm x 15 mm, cat. No. FB0875713 Fisher Scientific, USA) within a 24-hour period were collected from the outbred population with balanced sex ratios by slicing out a thin surface layer of media anchoring 50–90 eggs estimated visually and grafting onto a vial surface (25 mm x 95 mm, Polystyrene Reload, cat. No. 32109RL, Genesee Scientific, USA) containing a treatment diet for development on either HP or LP diet. Emerging flies (12–14 days post-oviposition, po) were released randomly into each of eight large cages (dimensions 20.3 cm x 20.3 cm x 20.3 cm) such that flies emerging from the HP and LP larval diets are each split equally and reciprocally allocated to each of the same diets in adult treatments (Fig 1). The final number of flies (N) in HH, HL, LH and LL treatments was 638, 676, 856, and 742, respectively. Each cage received a fresh

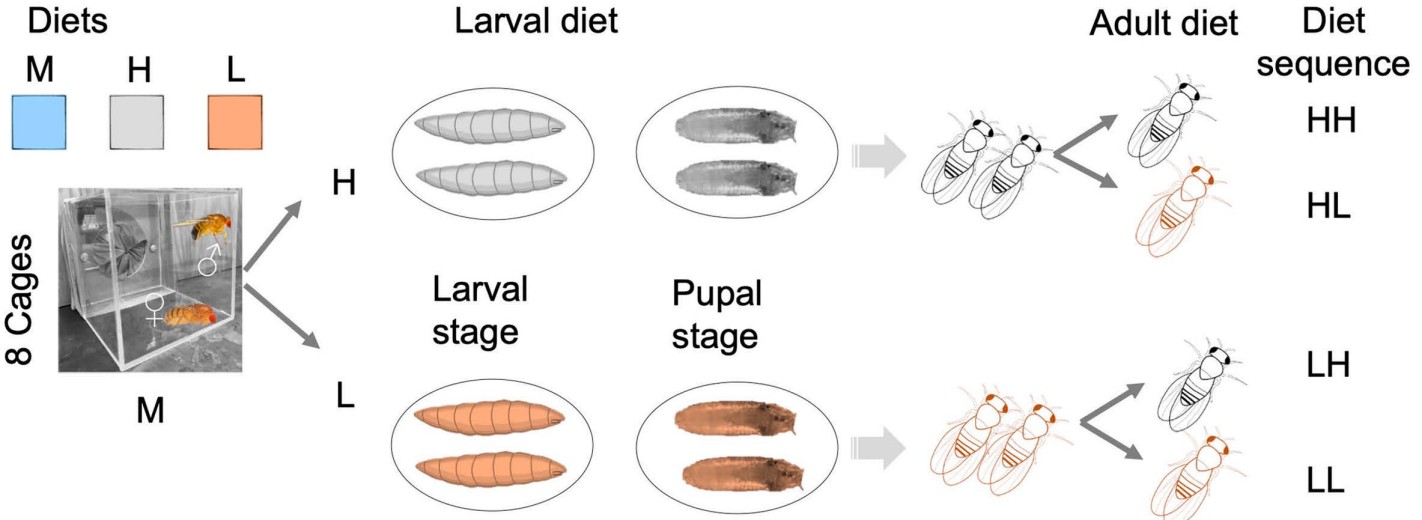

**Fig 1. A factorial study design.** Eggs collected from an outbred population reared on maintenance diet (M) were seeded on H and L diets for development. Emerged flies from each treatment were split to H and L diets as adults in two large cage replicates for each treatment ($N_{CAGES}=8$). From each cage we measured fecundity and lifespan.

plate of food three times a week (Monday, Wednesday, Friday). In addition, each cage received a separate micro-plate (60 mm x 15 mm, cat. No. FB0875713A, Fisher Scientific, USA) containing moist cotton wool as an additional source for drinking.

## Diet description

Experimental larvae and adults were raised on food manipulated mildly for protein levels to create H (~48%) and L (~38%) protein diets. Thus, the LP diet contained ~0.8x yeast concentration relative to the HP diet with 1x. The DSPR RILs and the derived outbred population were reared on a cornmeal-dextrose maintenance diet (M, ~10% protein). Detailed estimates of nutrient content of these diets can be accessed in [38]. Therefore, our goal is to compare fitness responses between assay treatment diets only and not their ancestral M diet. In all diets we used inactivated SAFPro Relax+YF 73050 yeast brand (Lesaffre Yeast Corp., Milwaukee, USA). All prepared diets were stored at 4°C and used within 14 days of preparation. Each cage received a fresh plate of food every Monday, Wednesday, and Friday, and with each egg quantification event. We reared experimental flies in a temperature-controlled chamber at 23°C, 24:0 light-dark cycles, which are typical rearing conditions for the DSPR.

## Phenotype measurement

We measured two life history phenotypes: lifespan, as number of days an individual lived from eclosion until death or censoring, and fecundity, as the number of eggs oviposited in a 3-hour period, three times a week (Monday, Wednesday, and Friday) consistently at the same time of day (12 noon to 3 pm). To measure lifespan, we counted the number of dead flies in each cage by sex, noting censored individuals (escaping or killed accidentally) whenever possible. Food plates bearing eggs for fecundity estimation were stored in a −20°C freezer until enumeration using approaches described in [38,39]. Briefly, we filtered eggs from the media onto black fabric discs using a custom vacuum suction pump and took images in a light-controlled chamber with a Canon Rebel TS*i*, (Canon Inc, Japan) camera at exposure time 1/25 s, aperture F5.6, and ISO 100. Eggs on one disc represented collective fecundity of females in a single cage for a period of 3 hours. We used

ImageJ software [40,41] to count eggs from each image manually by clicking over eggs and the tally exported to a comma separated value (csv) file. Using the record of death and censoring on each phenotyping date, we calculated average age-specific female fecundity three times a week until survival reached zero.

## Statistical analysis

**Survival.** We analyzed survival under four larval-adult nutritional regimes (HH, HL, LH and LL), each representing a unique interaction between larval and adult diets. To assess survival patterns, we employed time-to-event survival models and conducted log-rank tests using the R Survival package [42,43]. We further characterized longitudinal survival trends by examining specific survival percentiles—90% (capturing early, biologically driven mortality patterns), 50% (the median), and 10% (representing maximal longevity as it approximates the mean lifespan of the longest-lived 10% of a cohort [44,45].

Our exploratory analyses revealed temporal variations in survival rates influenced by treatment and sex. To determine whether the risk of adult mortality varied with larval diet, we constructed a Cox proportional hazards model that incorporated larval-adult diet, sex, and cage as covariates. As the proportional hazards assumption was violated for larval diet and sex, we stratified the model on sex, allowing male and female strata to have distinct baseline hazards while assuming constant coefficients across strata [46,47]. This adjustment resolved the violations, as indicated by a non-significant global test ($\chi^2 = 4.635$, $p > 0.05$) and for larval diet ($\chi^2 = 3.509$, $p > 0.05$). Attempts to stratify by both sex and larval diet led to small strata and non-informative errors, as expected (see Gordon, [46]). Subsequently, we constructed full models for three datasets: 1) both sexes combined, 2) females only, and 3) males only. For each dataset, we evaluated a comprehensive set of sub-models using the dredge() function from the R Package **MuMIn** [48]). We analyzed the following models:

1. Age ~ Larval_Adult * strata(Sex) + Larval_Adult *Cage

2. $Age_{Females}$ ~ Larval_Adult * Cage

3. $Age_{Males}$ ~ Larval_Adult * Cage

We selected models based on model ranking using the Akaike information criterion for smaller samples (ΔAICc and associated Akaike weights). Where models tied within 2 points of ΔAICc, we summarized averaged competing models using the model.avg() function in MuMIn. Whenever models were averaged, we report on conditional averages only (i.e., model set averaged).

**Fecundity and fitness.** We examined fecundity data in several ways. First, for each larval-adult diet regime, we calculated and visualized lifetime 1) group, and 2) per female fecundity. Next, we calculated age specific fecundity from daily per female fecundity. We used dredge() to evaluate all possible models from the following linear model:

Eggs ~ Larval_Adult*Age + Larval_Adult*Cage.

Next, we searched for patterns of sustained high and low egg output periods across a time series using the **change-point** R package [49]. Then, we compared fecundity patterns with either lifespan quantiles (25th, 50th, and 75th), or traditional survival categories: early (90%), median (50%) and maximum lifespan (10%). Lastly, we calculated a proxy of fitness for females in each of the four treatments (HH, HL, LH and LL) following the reasoning in Matthews et al, [50]. Since we enumerated deaths and eggs three times a week, we had a mix of two-day and three-day intervals. We thus computed age-specific survival as the mid-interval survivorship for individuals from age $x$ to $x+n$ days ($_nL_x$). We calculated age-specific fecundity ($m_x$) at each interval as number of eggs in the interval ($b_x$) divided by the number of females surviving, $k_x$ (i.e., $m_{x} = b_x/k_x$). We then used $_nL_x$ and $m_x$ as input to construct Leslie matrices with assumptions as in Mathews et al, [50] for each diet treatment. In the matrix, fecundity values form the top row while corresponding survival values line the diagonal, allowing to calculate the asymptotic population growth rate (lambda, λ). Lambda is mathematically the dominant eigen value of the projection matrix and can be interpreted as an approximate measure of fitness in a stable age distribution

[51]. In this study where we examine a single generation, we calculate a reproductive value ($V_x$), the average fecundity of an individual expected at a given age as a measure of age-specific fitness. We used the function eigen.analysis() in the R package **demogR** [52] to calculate λ. With each reproductive value, we also extracted rho – a measure of the rate by which the population is expected to converge asymptotically to a stable distribution with a rate at least as fast as log(rho).

## Results

We analyzed patterns of survival and fecundity in eight populations of fruit flies reared in two diet conditions as larvae and split to each of the same diets as adults in two replicates, N > 300 individuals per replicate (Fig 1). We calculated adult survival probability, lifetime and per female fecundity, and a measure of fitness derived from combining lifespan and fecundity values at each sampling date. Our goal was to assess the combined effect larval and adult diet regimes on the trajectories of these traits. Therefore, we fit models with coupled larval and adult diet terms while controlling for sex and cage (Table 1). Overall, we observed four major patterns in survival trajectories: 1) regime-dependent effects, 2) complex sex effects, 3) overall survival benefits when both larval and adult diets were LP, and 4) greater differences in post-median life phases, especially in males. Fecundity was overall 1) higher when larval diet was LP, but the timing of egg laying shifted – advanced in adult HP and delayed in adult LP treatments(Table 1). In addition, substantial fecundity was observed in older post-median flies (> 50 days) in most treatments. These patterns appeared independent of larval-adult diet sequence.

**Survival patterns depended on diet regime.** Overall differences in survival among the four diet treatments revealed a significant difference in survival curves (log-rank test, $\chi^2 = 47.1$, df = 3, p < 0.0001, S2 File). Pairwise comparisons using the log-rank test with Benjamini–Hochberg correction for multiple testing identified specific treatments that differed significantly. First, HL differed significantly from both HH (p = 1.6e-07) and LL groups (p = 6.6e-06). Second, LH differed significantly from the HL (p = 4.8e-06) and LL groups (p = 0.00018). No significant difference was observed between the HH and LH groups (p = 0.216).

**Table 1. Summary of survival, lifetime, and per female fecundity in each larval-adult treatment group.**

| Regime | Survival | All | Female | | Male |
|---|---|---|---|---|---|
| | rate[‡] | Days (CI) | Days (CI) | Eggs±se* | Days (CI) |
| HH | 0.9 | 10(10-12) | 12(10-15) | 6.5±1.6 | 8(8-12) |
| | 0.5 | 31(29-33) | 26(26-29) | 3.9±0.6 | 38(36-40) |
| | 0.1 | 61(57-66) | 57(57-61) | 1.7±0.4 | 68(64-78) |
| | | | | 3664[ltf] | |
| HL | 0.9 | 12(10-15) | 15(10-17) | 6.8±1.1 | 10(8-19) |
| | 0.5 | 38(36-40) | 36(36-40) | 8.5±2.1 | 38(36-40) |
| | 0.1 | 71(68-78) | 64(61-68) | 1.0±0.2 | 87(85-92) |
| | | | | 3432[ltf] | |
| LH | 0.9 | 12(10-15) | 12(12-15) | 12.1±2.6 | 10(8-17) |
| | 0.5 | 36(33-36) | 31(29-33) | 5.1±1.1 | 38(38-40) |
| | 0.1 | 68(64-71) | 64(59-66) | 1.0±0.2 | 78(71-78) |
| | | | | 9795[ltf] | |
| LL | 0.9 | 12(12-15) | 12(12-15) | 6.8±0.6 | 12(10-19) |
| | 0.5 | 38(36-40) | 36(33-38) | 12.6±2.4 | 43(40-47) |
| | 0.1 | 68(66-73) | 64(59-66) | 1.2±0.2 | 87(78-92) |
| | | | | 5533[ltf] | |

[‡]Survival rate is the number of days for that group to reach 90%, 50% and 10% survival; CI, confidence intervals around survival estimates; *mean number of eggs per female per 3 hour-laying period and standard error of mean fecundity (se), *; *ltf*, lifetime fecundity.

Survival was lowest in HH flies overall, with a median of 31 days (Table 1, S2 File), suggesting that constant higher protein from larval to adult age limited survival. Compared to HL and LL, the LH curve showed increased mortality after median survival, like that seen in HH flies (S2 File), suggesting that lower developmental protein followed by higher adult protein is a less optimal sequence for longevity. We note however that this sequence improves survival in the early half of lifespan compared to HH. The fact that HL and LL incurred the least mortality rates suggest that 1) adult diet, especially if it is LP, has a larger contribution to longevity, and 2) that survival is not more dependent on diet sequence as it is on potential interaction of the sequence. These results suggest that specific combinations of larval and adult nutritional treatments influence survival, with some treatments showing more pronounced differences than others.

**Survival was sex-dependent with variable dependence on diet sequence.** Exploratory analysis suggested a large effect of sex, in addition to diet regime effects (summarized in Table 1, S2 File). An overall log-rank test comparing survival curves across the eight groups defined by larval treatment, adult treatment, and sex showed highly significant differences ($\chi^2 = 146$, df $= 7$, p $< 2e–16$). The majority of post-hoc pairwise comparisons (Benjamini–Hochberg adjusted) showed highly significant contrasts (only 7 of 28 comparisons were not significantly different indicating strong effect of sex in shaping survival outcomes. The two shortest median survival occurred in HH and LH females while the longest median was in the LL diet sequence in males. Similarly, maximum lifespan was recorded for males in LL and HL combinations tied for the longest-lived groups at 87 days. We examine these differences more closely next.

### Median survival

The shortest median survival appeared in HH females consistently higher protein-fed (26 days), and LH females, fed higher protein only in adult stage (31 days). The longest median survival was observed in males on a constant low LL regime (43 days). To get a sense of the extent of sex differences as a function of larval diet, we visualized survival for early H vs early L for sexes separately and summarize the major patterns here (Fig 2, S2 File, Table 1):

- Adult L diet increased female median survival by 38.5% (i.e. by 10 days, from 26 days in HH to 36 days in HL.
- Males reared under all conditions tied at 38 days in median lifespan, except in LL (43 days), essentially ruling out effects of diet sequence in males. However, this effect also appeared in HL and LL females (tied at 36 days).
- A sizable contrast in the median (i.e. 19.4%) appeared between males and females constantly fed L, with males attaining the longest recorded median lifespan of 43 days (vs 36 days in females) emphasizing differential sex-based responses.
- That the 10-day lifespan improvement in HL over HH females is maintained in LL females suggests that larval L was not unnecessarily stressful if the L treatment was maintained in adults. Nonetheless, when larval diet was L, median lifespan is modestly shorter for female LH compared to LL (31 days vs 36 days) in favor of a constant L regime. Importantly, the LL lived 5 days longer in median lifespan over the LH regime in both sexes.

Together, these results indicate that combined effects of larval and adult nutritional treatments pervasively interacted with sex and significantly shaped survival outcomes.

### Maximum survival

Major improvements in maximum lifespan occurred in all regimes relative to the constant HH treatment demonstrating longevity benefits of the L diet overall, irrespective of diet sequence (Table 1, Table 1 in S1 File). The smaller gains accruing to LH in overall and male maximum survival signify the long-term cost of a low-to-high diet transition although these still represent large increases over HH. Notably, males in all treatments gained significantly in maximum lifespan (78–87 days) over HH females (57 days). Females in HL, LH, and LL tied at 64 days possibly indicating no further gain in days was induced by the L diet beyond that possible by an LH interaction. The largest male survival gains in LL and HL of 19 days over HH (68 vs 87 days) confirm that the longevity effects of a constant LL regime can be reached by an H diet

if a synergistic interaction occurred with an L diet. Indeed, the gain of 9 days (78–87 days) in LL males over LH males underscore the positive effect of LL over HH in maximum male lifespan. This larval L life extension effect is also apparent in females, but much smaller (7 days). Like median survival, maximum survival patterns are less clearly aligned with diet sequence, although the L diet consistently leads to increased longevity.

### Median to post-median effects were mixed across larval treatment groups

Major adult survival differences involving rank changes generally occurred post-median (S2 File). Further, larger variability around the median phase occurred in females and post-median in males in both larval treatments (Fig 2). This pattern suggests delayed effects of regime x sex interaction in males. For example, there was a 30-day difference between median and maximum lifespan of males within the HH treatment, compared to 49 days in the HL (Table 1 in S1 File). We interpret the fact that HL males lived 19 more days in post-median lifespan than HH males as an interaction effect of a diet switch to L.

### Effect of diet sequence on mortality risk

To further understand the effect of larval-adult diet sequence on survival, we fit a Cox model containing all major terms (i.e. Age~L_A * s(S) + L_A*C) and systematically evaluated all sub-models for combined male and female data (see methods). Performed on all data we found a significant effect of cage, with several models tied in ΔAICc weights (Table 2,

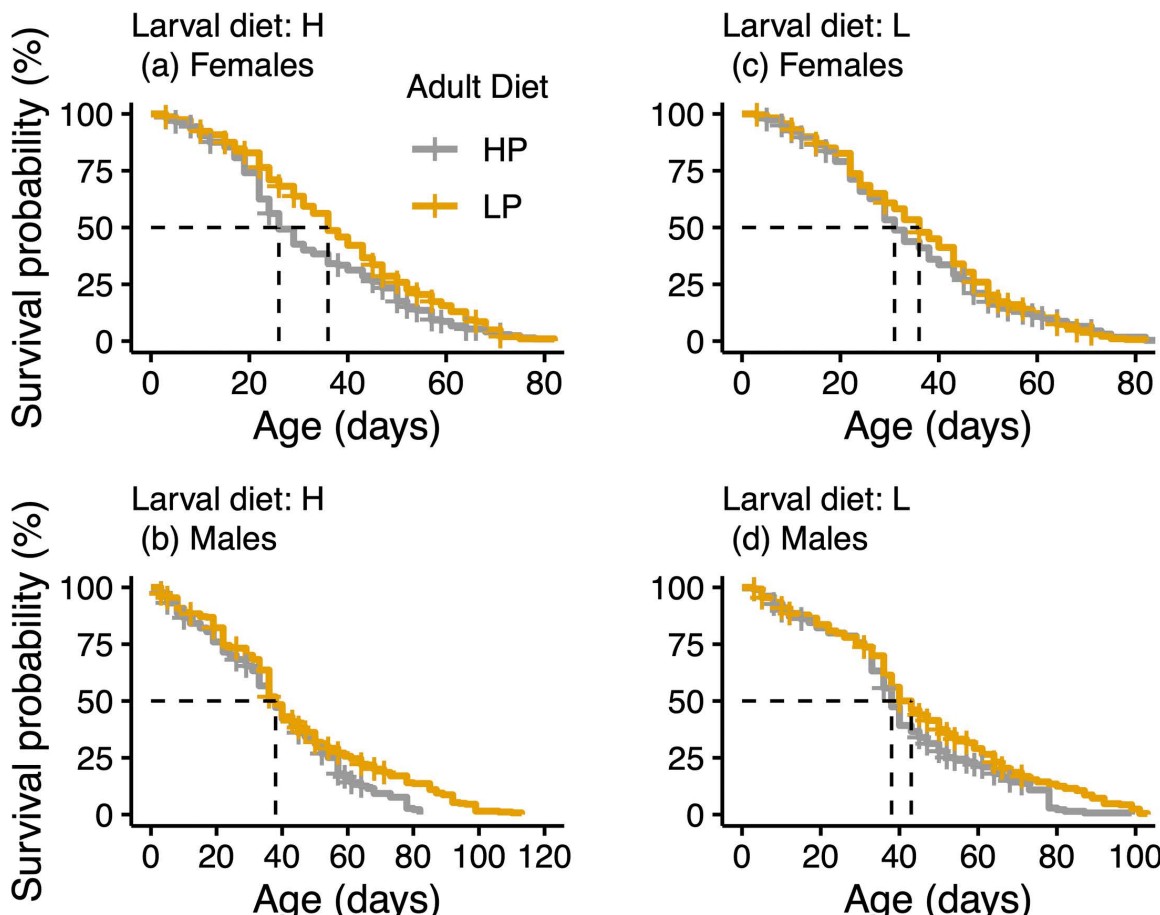

**Fig 2. Probability of survival grouped by larval-adult diet regime and sex.** Broken lines depict age (days) at median survival probability. Survival responded in an age-specific manner to treatment combinations, and differently between the sexes.

Table 2 in S1 File), and no effects of diet in conditional averages (Table 3 in S1 File). A marked effect of cage might in part arise from the fact that there were generally twice as many female events compared to male events except for HL (Table 1 in S1 File). To address this imbalance, and because exploratory analysis suggested a big role of sex, we repeated the model on each sex separately (Table 3 in S1 File). For females, several models were tied within 2 AICc points. Model averaging identified significant interaction effects of larval-adult diet in HL ($p<0.001$), LH ($p=0.0487$), and LL ($p=0.0037$) relative to HH (Table 3 in S1 File). For males, one model containing only the interaction term (L_A) was identified showing significant effects for two diet sequences, HL ($p<0.001$) and LL ($p<0.001$), relative to HH. These results highlight a significant role of larval-adult diet interactions in driving mortality risk in this population. Switching from low larval to high adult protein may be potentially detrimental on lifespan. Detailed model results are presented in Table 2 in S1 File.

## Low larval protein improved lifetime fecundity

We counted the number of eggs laid on media in each cage in a 3-hour period, three times per week over lifetime (N = 8 cages). These are same cages in which lifespan was measured, allowing us to relate fecundity to lifespan performance within replicate. Regime did not have a statistically significant effect on lifetime fecundity (one-way ANOVA, F(3, 4) = 2.167, p = 0.235). LH had the highest mean fecundity (4897 eggs), followed by LL (2766 eggs), HH (1832 eggs), and HL (1716 eggs). The data met normality assumptions (Shapiro-Wilk test, p = 0.09538) and homogeneity of variances (Bartlett's test, p = 0.7185), supporting the ANOVA results. In a 3-hour period an LH female on average laid 1.2 eggs, compared

**Table 2. Model comparison.**

|  | Models | k | AICc | ΔAICc | Wt |
|---|---|---|---|---|---|
| LIFESPAN | **1)**All flies ($n=3134$; events = 2912) |  |  |  |  |
|  | s(Sex) + C | 7 | 37446.42 | 0.00 | 0.28 |
|  | L_A+s(Sex) + C | 7 | 37446.42 | 0.00 | 0.28 |
|  | L_A+s(Sex) + C+L_A*C | 7 | 37446.40 | 0.00 | 0.28 |
|  | 2) Females only ($n=1956$; events = 1843) |  |  |  |  |
|  | L_A+C | 7 | 24430.26 | 0.00 | 0.29 |
|  | L_A+C+L_A*C | 7 | 24430.26 | 0.00 | 0.29 |
|  | C | 7 | 24430.26 | 0.00 | 0.29 |
|  | L_A | 3 | 24431.65 | 1.39 | 0.14 |
|  | 3) Males only ($n=1178$; events = 1069) |  |  |  |  |
|  | L_A | 3 | 13023.54 | 0.00 | 0.63 |
| FECUNDITY | L_A+L_A*Age+C | 13 | 3489.9 | 0.00 | 0.44 |
|  | L_A+L_A*Age+C+L_A*C | 13 | 3489.9 | 0.00 | 0.44 |

Further model selection results are presented in S1 File.

**Table 3. Summary of reproductive values across four treatments.**

| Regime | λ | Max $V_x$ | Min $V_x$ | Range $V_x$ |
|---|---|---|---|---|
| HH | 2.576 | 6.384 | 0.200 | 6.184 |
| HL | 3.458 | 4.071 | 0.245 | 3.826 |
| LH | 3.546 | 6.773 | 0.209 | 6.564 |
| LL | 3.194 | 7.515 | 0.325 | 7.190 |

Lambda (λ) represents the population growth rate per generation, while the maximum and minimum reproductive values ($V_x$) indicate the range of individual contributions to future population growth. The range of $V_x$ shows the variability in fecundity across individuals within each treatment.

to <0.5 eggs in the rest, a 240% increase in fecundity (Table 2). This observation prompted us to assess effect sizes using Hedges *d* which suggested that moving from L to H favored higher fecundity (Fig 4). Indeed, LH showed stronger effects when compared to HL and HH respectively. We note that the shortest-lived group (HH) also showed much lower lifetime fecundity. More generally, lifetime and per female patterns are visualized in Fig 3 and S2 File. Overall, these results are consistent with the interpretation that low larval protein increased adult fecundity, although we cannot exclude alternative explanations such as diet-associated toxins or other unmeasured factors.

**Low-to-high protein transition favors higher earlier female fecundity**

We found significant differences in age-specific fecundity as a function of larval *vs* adult diet for two comparisons (LH vs HH, *p* = 0.013 and LH vs HL, *p* = 0.006, pairwise t-test). Although L generally increased overall fecundity

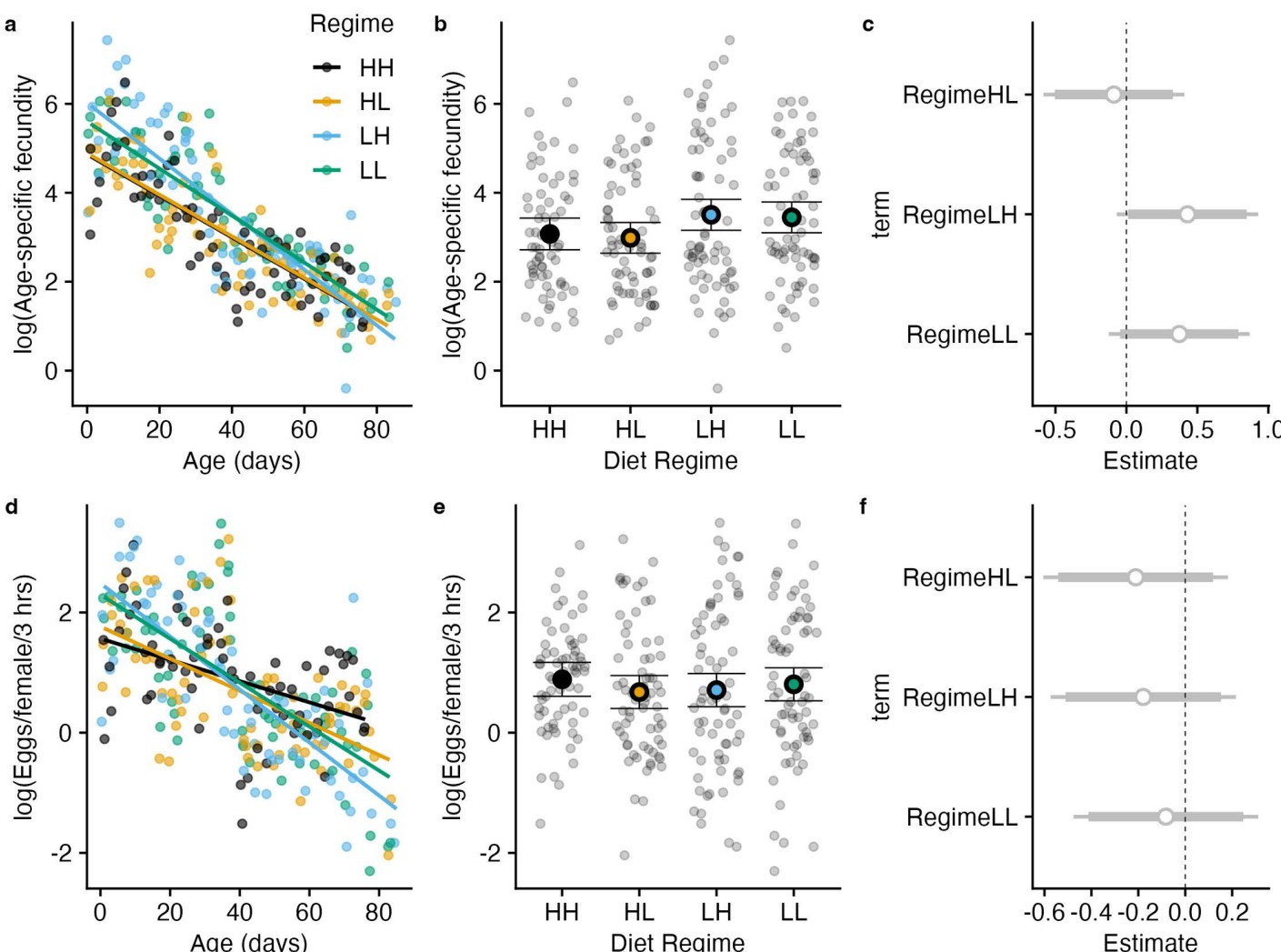

**Fig 3. Pattern of egg output in each treatment of larval-adult diet regime. a-c: Mean age-specific fecundity (i.e., mean number of eggs laid in a 3-hour period in each diet regime over lifespan is plotted.** Lines are linear models of those diets; **d-f:** same data in a-c calculated per female; **c,f:** are estimated regime effects calculated from regressing numbers of eggs on regime, ether in group fecundity **c** or per female fecundity f.

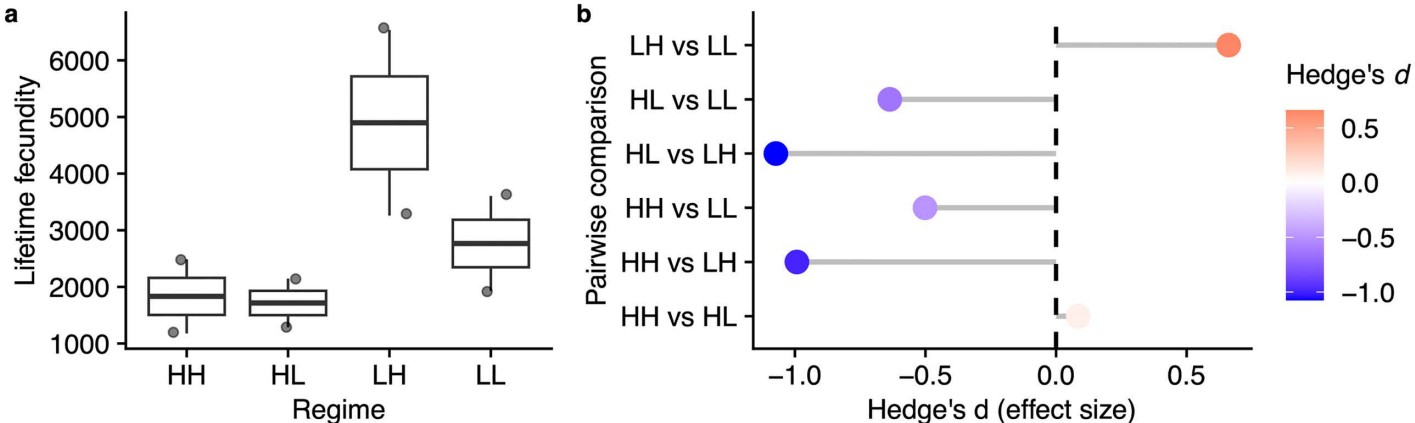

**Fig 4. Effect of regime on lifetime fecundity in *D. melanogaster*. Hedge's *d* statistic for pairwise regime comparisons.** Positive effects reflect a larger effect (mean) of the first diet sequence before 'vs' while negative effects reflect a larger effect of the second sequence. We interpret these results as $d < 0.2$ (small effect), $0.2 \le d < 0.5$ (medium effect), $0.5 \le d < 0.8$ (large effect), and $d \ge 0.8$ (very large effect) (Hedges and Olkin, 1985).

(Fig 3b), it did so in the pre-median life phase (Fig 5, note change in rank favoring larval H in the post-median phase). Notably, LH was 5 days early in median lifespan compared with both HL and LL likely accounting for higher advanced fecundity. Surprisingly however, this regime tied with HL in female maximum lifespan at 64 days suggesting within-regime variation in egg output vis-à-vis survival. While a longer lifespan provided more time for reproduction, heavy early fecundity seems to suggest little trade-off with lifespan in females of this treatment. We found a significant interaction effect of age with the LH regime (linear model est. −4.1, z = 4.02, p < 0.001). Finally, as expected, increasing age had a negative effect on age-specific fecundity overall (est. −2.5, z = 3.3, p < 0.001). Model results are reported in Table 3 in S1 File.

### Diet regime influenced timing and duration of peak fecundity

The age-specific pattern in fecundity revealed peak laying periods that were variably early or late but seemed unique to each regime, suggesting episodes or phases of egg laying. Although most eggs were expectedly produced earlier in life (15–38 days *po*), obvious patterns of prolonged dips and spikes were observed in this interval (S2 File). To gain further insight into this pattern, we looked for significant change points in the patterns of egg laying across lifespan in each treatment. We identified a pattern in which both regimes starting with larval L diet showed a longer initial period of sustained oviposition compared to treatments starting out with the H diet (Fig 5a, 5d) although the specific character of these episodes differs. Further, although fecundity plummets in all treatments after about 50 days, there remains substantial egg output, and in a few cases, some low-grade spikes in old females (S2 File).

### Female fecundity in the context of survival patterns

In this study, lifespan was lowest in a sustained high protein HH diet and tied for maximum lifespan in a low-low and a high-low regime in both median and maximum lifespan. Similarly, lifetime fecundity was lowest in a high-high regime but highest in a low-high treatment, resulting in a mild trade-off limiting the egg laying period in this group. Thus, a protein reduction generally increased both lifespan and fecundity in our lab conditions. However, in reproducing females, a switch from H to L favors lifespan more while that from L to H favors early reproduction more. To confirm this observation, we combined age-specific lifespan and fecundity values for each treatment into a fitness proxy using life tables and Leslie matrix calculations (see methods) and obtained a reproductive value ($V_x$) in each condition (Table 3, Fig 6).

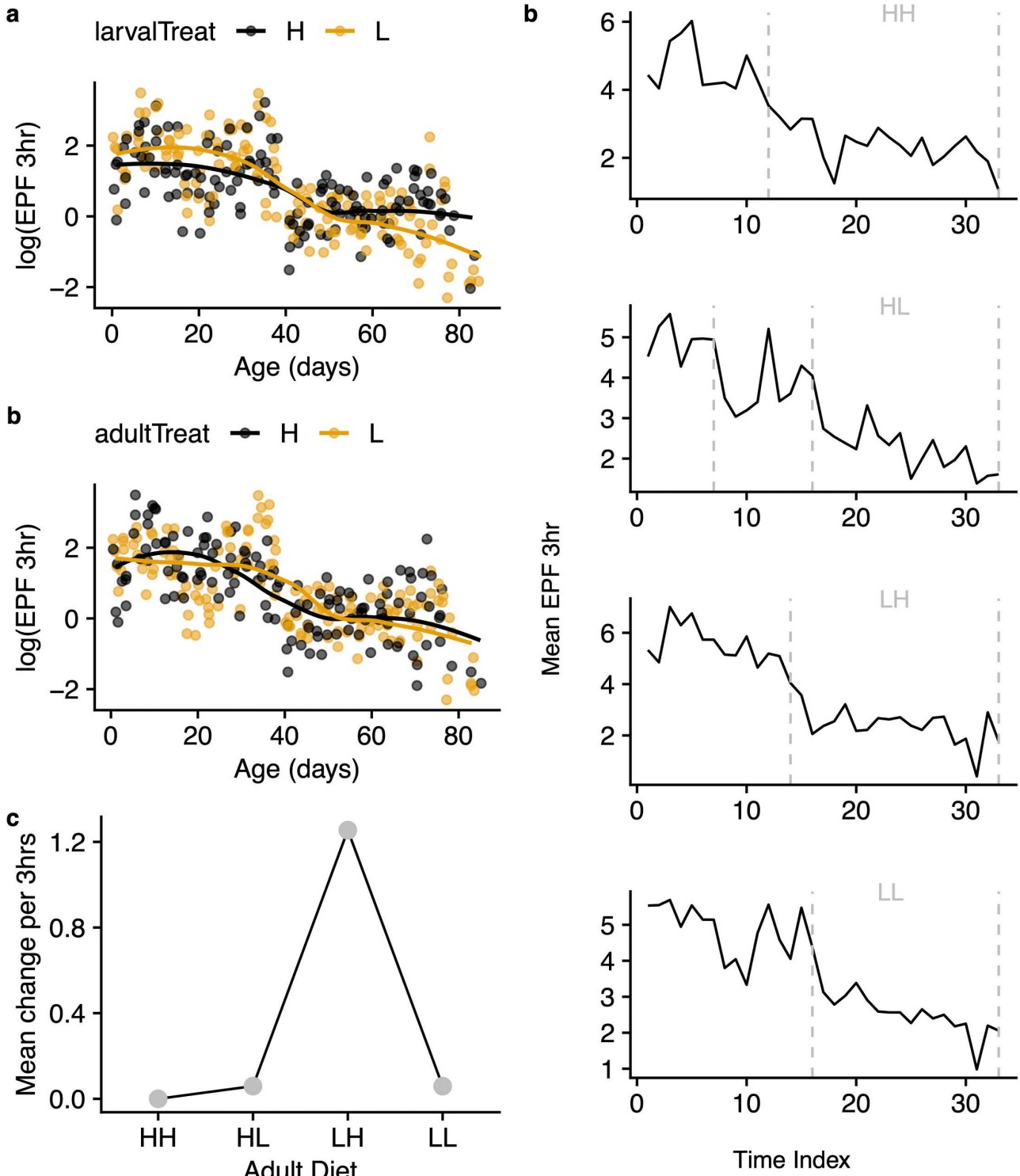

**Fig 5. Average number of eggs per female per day calculated from 3-hour laying periods sampled 3 times a week over lifetime.** *a* – grouped by larval treatment H or L. *b* – grouped by adult treatment H or L. *c* – Mean difference in egg number per 3-hour laying period in each regime. *d* – Number of detected change points in egg-laying and duration of these periods.

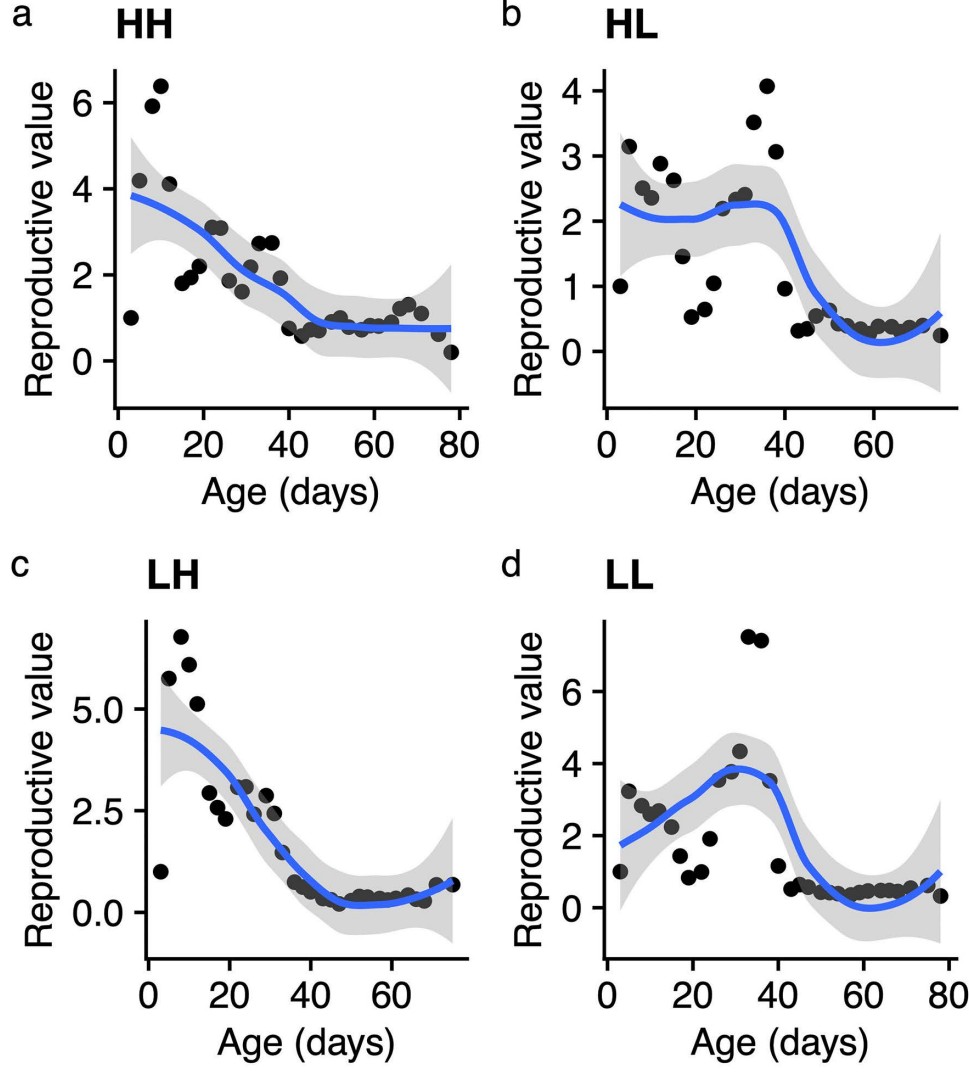

**Fig 6. Age-specific fecundity patterns in each of the four diet combinations.** A reproductive value is the expected per 3-hour fecundity of a female calculated from lifespan-fecundity Leslie matrices. The dots represent mean per female fecundity at each sampling time over the lifespan.

Reproductive value, a measure of the average individual female fecundity to be expected at that age, varied across treatments. The LL treatment had the highest maximum reproductive value (7.52), suggesting that females in this treatment retained high fecundity potential despite presumably adverse low protein conditions. LH and HH also demonstrated relatively high maximum $V_x$ (6.773 and 6.384, respectively), while the HL treatment had a lower maximum $V_x$ (4.071), indicating a potentially less efficient reproductive output despite higher growth rates (λ). The range of reproductive values also varied across treatments. The LL, LH and HH treatments showed wide ranges of reproductive values (Table 3), implying that there was considerable variation in reproductive success across individuals, suggesting that some individuals may have limited contributions to future generations. In contrast, the HL treatment exhibited a smaller range (3.826), indicating more uniform reproduction across ages. Comparisons of age-specific reproductive values between treatments are visualized in Fig 6. HH and LH decline almost monotonously before leveling out in old age. In HL and LL peak fecundity shifts to the right along the time axis representing a delay in their peak fitness performance. These trends generally hold when

fecundity data is analyzed on its own or in lifespan quantiles (S2 File), suggesting delayed but generally sustained reproductive output with adult L diet.

## Discussion

We studied fitness effects of early-adult nutritional experience in an outbred population generated from 835 recombinant inbred lines of a multiparent *D. melanogaster* population. Fitness in this study is a proxy obtained from full lifespan and partial fecundity (assayed three times weekly over lifespan in each treatment). We evaluated the combined effect of larval-to-adult nutritional experience on fitness proxy measures (lifespan, fecundity and reproductive value). We find that lower protein nutrition enhanced both lifespan and fecundity relative to a constant high protein diet irrespective of diet sequence in a lab environment. We further find that high adult protein produces an earlier peak reproductive period while lower adult protein delays the peak laying by about two weeks.

### Diet sequence was decoupled from fitness performance

Our L diet represents approximately a 21.4% protein dilution (see Table 1 in S1 File in [38]). We speculate that this moderate dilution did not impose significant nutritional stress and may have been buffered by plastic responses in our population. Previous research on adaptive responses to nutritional challenges in an experimental evolution context found that a lower-calorie developmental diet led to faster development but reduced adult weight [25]. However, morphological changes such as ovariole size or number, which have been linked to dietary variation in some studies (e.g., Deas et al. [4]), are unlikely to have played a major role in our system. This is because we reared our flies under relatively low larval and adult densities, potentially mitigating strong selection pressures on reproductive traits within a single generation. Our findings align with a study in the African clawed frog, which followed a similar experimental design [53]. In that study, manipulating dietary protein influenced hormonal effects on male vocal ability, yet the responses were mixed, suggesting that diet-induced phenotypic changes may be complex and context-dependent.

### Low protein larval diet improved fitness

Regardless of sequence, early L diet was associated with higher lifetime fecundity. This was true generally for lifespan also, except for the 5 days reduction in median lifespan in the LH, which also showed 2–3 times more eggs as other regimes. This regime also advanced fecundity in producing an early peak (Fig 5). We therefore interpret lower median survival in LH as arising from a steeper survival-fecundity trade-off. But why would a relatively low protein diet favor increased fitness? It is conceivable that significant resources are allocated to locomotor activity and enhance functional resilience. A study in *D. melanogaster* in which in addition to fecundity and lifespan, measured activity level also in a series of protein restricted diets (30–90% protein), concluded that chronic protein restriction in all diets from eclosion did not induce malnutrition [54]. Thus, apart from the key lifespan-reproduction trade-off, high fecundity in L treatments might relate to improved locomotor function and generally a slower pace in functional senescence [55]. With slower declines in activity, L flies may engage in more reproductive activity longer compared with those on high protein diet. However, we note that while L treatments led to higher fitness due to a longer egg laying period, the timing of fecundity is critically important for fitness in wild populations, and early fecundity is likely a major benefit in wild *D. melanogaster.*

As for why low protein in development should result in high fecundity, we speculate that deficits in development were potentially compensated for by a significant delay in peak laying as adults (i.e. LH regime). Where this compensation is limited, a cost is paid in median lifespan (i.e. 5 days lower in LH). In the bank vole (*Myodes glareolus*), Schroderus et al, [56] could not explain evident phenotypic trade-offs in offspring number *vs* size by a genetic trade-off, but rather by negative correlations in environmental effects. In that study, some genetic correlations estimated were positive leading them to conclude that genetic variation in those traits may not always be antagonistic. Our results may underpin potential non-genetic environmental effects.

## Sex-based patterns

We observed sex-specific differences in survival and reproductive output, particularly in the post-median lifespan phase. While both sexes exhibited long post-median lifespans, males showed pronounced rank shifts in survival trajectories (see S2 File). Egg production followed stage-specific rather than strictly age-dependent patterns, suggesting different underlying response mechanisms to dietary combinations. Simulation studies [57] indicate that environmental perturbations can shape demographic rates differently, potentially leading to divergent age- *vs* stage-specific patterns, particularly when strong survival-fecundity trade-offs exist (e.g., LH regime).

Males generally survived longer than females, independent of diet treatment, with the greatest sex difference in maximum lifespan. Several factors could explain this pattern. From an energetics perspective, sexual size dimorphism influences sensitivity to environmental conditions, with larger-bodied individuals (typically females) at greater risk of mortality under low food availability [5,58,59]. While this dimorphism may not be evident at the larval stage, females' higher energetic demands in adulthood may increase their mortality risk. Furthermore, sex differences in foraging behavior may contribute to survival differences. Davies et al. [58] found that adult females maintained a consistent protein-to-carbohydrate (P:C) ratio, whereas males adjusted their intake to compensate for developmental deficiencies. In conditions lacking dietary choice (as this study), lifespan increased as P:C ratio decreased, independent of larval diet. Another possible explanation might involve male density effects. While high male density in *Bicyclus anynana* butterflies was linked to increased body mass and energy storage, ultimately enhancing survival under competition [60], we controlled for density in this study. Both during egg collection and within experimental cages, population densities remained well below maximum capacity (i.e., ~600–900 flies in cages designed for >3000).

Finally, male presence itself may contribute to female mortality. While we did not control sex ratios, male-male competition may indirectly stress females in a closed cage space. Bretman et al. [61] showed that mere exposure of female *D. melanogaster* to males accelerated functional aging, increasing mortality risk independent of body mass, fat storage, or sex peptide changes. These findings align with our results, suggesting that male-driven stress may play a role in sex differences in lifespan.

## Complexity of nutritional effects on life history traits

In recent years, several studies have examined the effects of developmental nutrition on resulting life history traits [25,31,62]. May et al, [62] found that both rich and poor diet imposed suboptimal conditions for development and adult size, yet increased virgin adult lifespan. Manipulations of reproductive potential revealed environment-dependent shifts in the timing of reproduction. In a later study, the same investigators [28] showed that that diet mismatches were not necessarily detrimental to lifespan or fecundity, with adult diet exerting greater influence. Experimental evolution further demonstrated that responses to diet depend on the sequence of larval and adult environments, and vary across life stages [25]. Similarly, Duxbury et al [31] reported complex, sex-specific nutritional effects on lifespan trajectories.

Our study largely confirms these findings and adds a major observation: a mildly protein-restricted larval diet increased both lifespan and fecundity. We speculate that these outcomes reflect differences in the genetic background of the populations studied. For instance, the Dahomey population used by Duxbury et al. [63] originated from a single West African collection, while the laboratory stock (S) population examined by May et al. [62] represents regional European diversity. By contrast, our test population was derived from the Drosophila Synthetic Population Resource, which captures a broad and globally distributed pool of genetic variation. This diversity may explain why we observed patterns not consistently seen in other laboratory lines, including greater maximum lifespan in males. Although many studies report female-biased longevity in *Drosophila* [64–66], sex differences vary across genotypes and environments, with some strains showing male-biased or no clear difference [67–69]. Taken together, these comparisons underscore that sex- and stage-specific responses to nutrition are highly context-dependent, shaped by both environmental inputs and the underlying genetic architecture of the population.

## Conclusions

We investigated the combined effects of larval and adult diets that differed only in protein quantity. Our test population, outbred from a diverse set of inbred lines derived from eight global founder lines, provided a genetically heterogeneous framework for assessing fitness outcomes.

Our findings reveal a major role of larval-adult diet interaction effects in the determination of both lifespan and fecundity. Interestingly, our population suggest that reduced protein availability enhanced both fitness components in a strongly sex-specific manner. Specific strategies for fitness improvements in low protein environments appear to be shifting periods of peak fecundity as well as spreading reproduction over a longer period into later post-median ages. Future studies exploring adult trait expression should account for ontogeny-wide nutritional influences, as early-life conditions may shape later-life fitness in nuanced and context-dependent ways.

## Supporting information

**S1 File. Detailed survival and fecundity model results – tables.**
(DOCX)

**S2 File. Additional survival and fecundity results – figures.**
(DOCX)

## Acknowledgments

Anna Perinchery-Herman constructed the lighting studio box we used for image capture. Several King Lab members performed tasks at different times that supported this work.

## Author contributions

**Conceptualization:** Enoch Ng'oma.

**Data curation:** Andrew Jones, De'anne Donnell, Elizabeth G. King, Enoch Ng'oma.

**Formal analysis:** Andrew Jones, Enoch Ng'oma.

**Funding acquisition:** Elizabeth G. King, Enoch Ng'oma.

**Investigation:** Andrew Jones, De'anne Donnell, Elizabeth G. King, Enoch Ng'oma.

**Methodology:** Andrew Jones, Elizabeth G. King, Enoch Ng'oma.

**Project administration:** Elizabeth G. King.

**Supervision:** Elizabeth G. King, Enoch Ng'oma.

**Validation:** Andrew Jones, De'anne Donnell, Elizabeth G. King, Enoch Ng'oma.

**Visualization:** Andrew Jones, Enoch Ng'oma.

**Writing – original draft:** Enoch Ng'oma.

**Writing – review & editing:** Andrew Jones, De'anne Donnell, Elizabeth G. King, Enoch Ng'oma.

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
