## [Decision Letter · Decision Letter 0]

8 Aug 2025

PONE-D-25-27356

Interactive effects of developmental and adult nutrition on lifespan and fecundity in a genetically diverse Drosophila population

PLOS ONE

Dear Dr. Ng'oma,

Thank you for submitting your manuscript to PLOS ONE. After careful consideration, we feel that it has merit but does not fully meet PLOS ONE’s publication criteria as it currently stands. Therefore, we invite you to submit a revised version of the manuscript that addresses the points raised during the review process.

Based on the referee’s comments and my own review, the paper needs to be revised to clarify some of the key aspects of the study before it can be published. Please address all the comments raised by the reviewer. 

We look forward to receiving your revised manuscript.

Kind regards,

Barbara Jennings

Academic Editor

PLOS ONE

Journal Requirements:

[This work was supported by a NIH grant R01 GM117135 to EGK, and the University of Missouri startup funds to EN.].

Reviewers' comments:

Reviewer's Responses to Questions

**Comments to the Author**

1. Is the manuscript technically sound, and do the data support the conclusions?

Reviewer #1: Yes

2. Has the statistical analysis been performed appropriately and rigorously? 

Reviewer #1: Yes

3. Have the authors made all data underlying the findings in their manuscript fully available?

Reviewer #1: Yes

4. Is the manuscript presented in an intelligible fashion and written in standard English?

Reviewer #1: Yes

5. Review Comments to the Author

Reviewer #1: The paper investigates how larval and adult diet interact to influence lifespan and fecundity in a genetically diverse population of Drosophila melanogaster. Relatively few studies have

explored how interactions between developmental and adult diets affect fitness in genetically diverse populations. The authors investigate how developmental diet and adult nutrition jointly influence lifespan and fecundity in outbred Drosophila. While there are some interesting findings, some parts of the paper could be clearer. Some suggestions are below:

In the cartoon that represents set up with cages and foods, Fig1, were cages mixed cages with both males and females in one cage? Could they add fly sex symbols to represent if the males and females were housed together?

Line 95-97, unclear. Please re-write. It says: adult diet was determinant of survival, however in the next sentence it is stated that developing on high-yeast diet benefited adult lifespan regardless of adult diet? Unclear is adult diet or larval diet is determinat of survival?

Line 99-100. A bit unclear and how is this linked to Drosophila:

“Similarly, a rodent study indicated that diet-induced hyperphagia (overeating) is greater in males, but females display a higher preference for high-fat diets. “

Line 158: detailed died should be in this paper not in supplementary of ref 38.

Line 164 says 24:0 light-dark cycle. Please check if this was 12:12 light-dark cycles.

Were both exes housed together in a cage? How many flies were housed per cage and how many cages were there per treatment?

Line 253: Unclear here what is L in this context :” 3) overall survival benefits of both larval and adult L on both traits,”. I understand L is low protein food but it this sentence maybe better to clarify.

Line 259-266 refers to S2 File. Text refers to LH food whereas Fig S2 legend says C DR food. Could this please be corrected?

Line 380 it is stated that “Overall, these results suggest that our low larvae protein treatment increased adult fecundity. “ How can you exclude that there was not a toxin in the food and that lower protein diet meant lower levels of toxin lead to increased fecundity?

They observed that males survived longer than females. How common is this for Drosophila, please compare with some other Drosophila literature using different strains. To the best of my knowledge there are Drosophila strains where females are longer lived than males, and this may even be more common.

6. PLOS authors have the option to publish the peer review history of their article (what does this mean?). If published, this will include your full peer review and any attached files.

Reviewer #1: No

---

## [Author Response · Author response to Decision Letter 1]

21 Aug 2025

Response to reviewer

In the cartoon that represents set up with cages and foods, Fig1, were cages mixed cages with both males and females in one cage? Could they add fly sex symbols to represent if the males and females were housed together?

The reviewer is correct, fly cages had both females and males at roughly equal sex ratios. We have made the following revisions to the manuscript:

1. We mention explicitly in the methods that sex ratios in the 8 cages were balanced. The particular line in the methods now reads: “To obtain assayed individuals, eggs oviposited on media plates ….. within a 24-hour period were collected from the outbred population with balanced sex ratios by slicing out a thin surface layer of media anchoring 50 - 90 eggs estimated visually….”

2. We have added visual male and female thumbnails and sex symbols to Figure 1as suggested by reviewer.

Line 95-97, unclear. Please re-write. It says: adult diet was determinant of survival, however in the next sentence it is stated that developing on high-yeast diet benefited adult lifespan regardless of adult diet? Unclear is adult diet or larval diet is determinant of survival?

The authors agree with reviewer: The way it is written makes it sound contradictory, because one sentence says adult diet is the main determinant of survival, while the next seems to say developmental (larval) diet also determines survival regardless of adult diet. We understand that what Duxbury & Chapman (2019) showed is that a) adult diet largely determined survival in both sexes, and b) that developmental diet (larval diet) had an additional, sex-specific effect on females, enhancing lifespan and reproduction when they developed on a high-yeast diet, independent of their adult diet. We have therefore revised the section to read:

For example, a study in Drosophila found that mismatches between developmental and adult nutrition influenced female, but not male, reproductive success [31]. Survival in both sexes was primarily determined by adult diet, with longer lifespans on high-yeast adult food. However, in females, developing on a high-yeast diet conferred additional benefits to lifespan and reproductive success, regardless of the adult diet consumed.

Line 99-100. A bit unclear and how is this linked to Drosophila:

“Similarly, a rodent study indicated that diet-induced hyperphagia (overeating) is greater in males, but females display a higher preference for high-fat diets. “

We thank the reviewer for picking up this discord in logic that we missed. Indeed, the Drosophila example is about developmental vs. adult diet effects, while the rodent study we cited (Maric et al. 2022) is more about sex-specific feeding behavior, energy expenditure, and susceptibility to diet-induced obesity. Thus, the link to Drosophila isn’t fully spelled out, so it comes across as a sudden jump in logic. To make it clearer, we have reframed the rodent sentence as another example of how males and females differ in their responses to diet and specify how that relates to the broader theme (sex-specific nutritional sensitivity across species). A revised section now reads:

These patterns of sex-dependent dietary response are not unique to insects. In rodents, males show greater hyperphagia (overeating), whereas females are more inclined toward high-fat diets and are somewhat protected from rapid obesity and metabolic decline through higher energy expenditure

Line 158: detailed died should be in this paper not in supplementary of ref 38.

Although published as supplemental data to our earlier study, we think that these data are publicly available and accessible. We have therefore amended the line in question to read:

Detailed estimates of nutrient content of these diets can be accessed in [38].

Line 164 says 24:0 light-dark cycle. Please check if this was 12:12 light-dark cycles.

We confirm that the light:dark cycle was indeed 24:0. The DSPR was originally established and maintained under continuous light (24:0) in the founding studies (King et al. 2012a; King et al. 2012b). Our work follows this convention, as we and others have conducted most previous experiments with the DSPR under these conditions. We acknowledge that the DSPR is now maintained and distributed by the Bloomington Drosophila Stock Center, likely under the standard 12:12 LD cycle. Light regime is known to influence physiological and behavioral outcomes, so we note this difference as an important consideration when comparing across studies.

Were both exes housed together in a cage? How many flies were housed per cage and how many cages were there per treatment?

We addressed the issues mixed/single sexes and number of cages both in Fig 1 and in the text as outlined under the first query and annotated in the revised manuscript. The number of cages and treatments are further outlined in the caption to Figure 1. Here, we added the following statement to the methods section to clarify the value of N for each treatment:

The final number of flies (N) in HH, HL, LH and LL treatments was 638, 676, 856, and 742, respectively.

Line 253: Unclear here what is L in this context :” 3) overall survival benefits of both larval and adult L on both traits,”. I understand L is low protein food but it this sentence maybe better to clarify.

We thank the reviewer for pointing this out. Indeed, single letter abbreviations L and H were never introduced earlier. We have therefore clarified these by quoting them as LP and HP introduced in the methods under diet description. The corrected text now reads:

Overall, we observed four major patterns in survival trajectories: 1) regime-dependent effects, 2) complex sex effects, 3) overall survival benefits when both larval and adult diets were LP, and 4) greater differences in post-median life phases, especially in males. Fecundity was overall 1) higher when larval diet was LP, but the timing of egg laying shifted - advanced in adult HP and delayed in adult LP treatments(Table 1). In addition, substantial fecundity was observed in older post-median flies (> 50 days) in most treatments. These patterns appeared independent of larval-adult diet sequence.

Line 259-266 refers to S2 File. Text refers to LH food whereas Fig S2 legend says C DR food. Could this please be corrected?

This is corrected in S2 File.

Line 380 it is stated that “Overall, these results suggest that our low larvae protein treatment increased adult fecundity. “How can you exclude that there was not a toxin in the food and that lower protein diet meant lower levels of toxin lead to increased fecundity?

We thank the reviewer for this thoughtful comment. We agree that in principle, lower fecundity on high-protein food could reflect the presence of a harmful compound rather than protein level per se. Our wording “suggest” was intended to reflect this possibility. While we cannot entirely exclude the presence of a toxin, several points argue against this as the sole explanation. First, all diets were prepared from the same base components with protein manipulated by adjusting yeast concentration, a common and well-established method in Drosophila nutrition studies. Second, we observed consistent effects of larval protein across multiple life-history traits (both fecundity and lifespan), which aligns with previous work showing that yeast/protein levels modulate these traits. Third, the patterns we observed (higher fecundity with lower larval protein) are consistent with published findings in nutritional geometry and dietary restriction literature, rather than pointing to an idiosyncratic toxic effect. We have revised the manuscript to clarify that our results are consistent with low larval protein increasing adult fecundity, while acknowledging that alternative explanations (including unintended toxic effects of high protein) cannot be fully excluded. The conclusion statement now reads:

Overall, these results are consistent with the interpretation that low larval protein increased adult fecundity, although we cannot exclude alternative explanations such as diet-associated toxins or other unmeasured factors.

They observed that males survived longer than females. How common is this for Drosophila, please compare with some other Drosophila literature using different strains. To the best of my knowledge there are Drosophila strains where females are longer lived than males, and this may even be more common.

We thank the reviewer for raising this important point. Sex differences in lifespan in Drosophila melanogaster are known to vary across genetic backgrounds, environmental conditions, and even laboratory protocols (Hoffman et al 2021). In some widely used laboratory strains (e.g., Canton-S, Dahomey, w1118), females often live longer than males (Promislow et al. 1996; Piper & Partridge 2007; Magwire et al. 2004). However, there are also reports where males live as long as or longer than females, particularly under certain dietary conditions or in specific genotypes (Lin et al, 2023; Lints et al 2009,Hoffman et al, 2021). Given that our study used a genetically diverse, outbred population, it is not unexpected that male-biased survival emerged. We interpret this as reflecting the combined influence of genotype-by-environment interactions rather than as a fixed sex difference across D. melanogaster. We have revised the last section of the discussion to reflect this idea more clearly:

Our study largely confirms these findings and adds a major observation: a mildly protein-restricted larval diet increased both lifespan and fecundity. We speculate that these outcomes reflect differences in the genetic background of the populations studied. For instance, the Dahomey population used by Duxbury et al. [63] originated from a single West African collection, while the laboratory stock (S) population examined by May et al. [62] represents regional European diversity. By contrast, our test population was derived from the Drosophila Synthetic Population Resource, which captures a broad and globally distributed pool of genetic variation. This diversity may explain why we observed patterns not consistently seen in other laboratory lines, including greater maximum lifespan in males. Although many studies report female-biased longevity in Drosophila [64–66], sex differences vary across genotypes and environments, with some strains showing male-biased or no clear difference [67–69]. Taken together, these comparisons underscore that sex- and stage-specific responses to nutrition are highly context-dependent, shaped by both environmental inputs and the underlying genetic architecture of the population.

References for male-biased lifespan

Hoffman JM, Dudeck SK, Patterson HK, Austad SN. Sex, mating and repeatability of Drosophila melanogaster longevity. Royal Society Open Science. 2021;8: 210273. doi:10.1098/rsos.210273

Lin Y-C, Zhang M, Chang Y-J, Kuo T-H. Comparisons of lifespan and stress resistance between sexes in Drosophila melanogaster. Heliyon. 2023;9: e18178. doi:10.1016/j.heliyon.2023.e18178

Lints FA, Bourgois M, Delalieux A, Stoll J, Lints CV. Does the female life span exceed that of the male: a study in Drosophila melanogaster. Gerontology. 2009;29: 336–352. doi:10.1159/000213136

---

## [Editor Report · Decision Letter 1]

26 Sep 2025

Interactive effects of developmental and adult nutrition on lifespan and fecundity in a genetically diverse Drosophila population

PONE-D-25-27356R1

Dear Dr. Ng'oma,

We’re pleased to inform you that your manuscript has been judged scientifically suitable for publication and will be formally accepted for publication once it meets all outstanding technical requirements.

Kind regards,

Barbara Jennings

Academic Editor

PLOS ONE
---

## [Editor Report · Acceptance letter]

PONE-D-25-27356R1

PLOS ONE

Dear Dr. Ng'oma,

I'm pleased to inform you that your manuscript has been deemed suitable for publication in PLOS ONE. Congratulations! Your manuscript is now being handed over to our production team.

Kind regards,

on behalf of

Dr. Barbara Jennings

Academic Editor

PLOS ONE